# A Rationale for Hypoxic and Chemical Conditioning in Huntington’s Disease

**DOI:** 10.3390/ijms22020582

**Published:** 2021-01-08

**Authors:** Johannes Burtscher, Vittorio Maglione, Alba Di Pardo, Grégoire P. Millet, Christoph Schwarzer, Luca Zangrandi

**Affiliations:** 1Department of Biomedical Sciences, University of Lausanne, 1015 Lausanne, Switzerland; 2Institute of Sport Sciences, University of Lausanne, 1015 Lausanne, Switzerland; gregoire.millet@unil.ch; 3IRCCS, Neuromed, 86077 Pozzilli, Italy; vittorio.maglione@neuromed.it (V.M.); dipardoa@hotmail.com (A.D.P.); 4Department of Pharmacology, Medical University of Innsbruck, 6020 Innsbruck, Austria; schwarzer.christoph@i-med.ac.at; 5Institute of Virology, Campus Benjamin Franklin, Charité—Universitätsmedizin Berlin, Corporate Member of Freie Universität Berlin, Humboldt-Universität zu Berlin, and Berlin Institute of Health, 12203 Berlin, Germany; Luca.Zangrandi@i-med.ac.at

**Keywords:** Huntington’s disease, hypoxia, mitochondria, S1P, opioid, NMDA

## Abstract

Neurodegenerative diseases are characterized by adverse cellular environments and pathological alterations causing neurodegeneration in distinct brain regions. This development is triggered or facilitated by conditions such as hypoxia, ischemia or inflammation and is associated with disruptions of fundamental cellular functions, including metabolic and ion homeostasis. Targeting intracellular downstream consequences to specifically reverse these pathological changes proved difficult to translate to clinical settings. Here, we discuss the potential of more holistic approaches with the purpose to re-establish a healthy cellular environment and to promote cellular resilience. We review the involvement of important molecular pathways (e.g., the sphingosine, δ-opioid receptor or N-Methyl-D-aspartate (NMDA) receptor pathways) in neuroprotective hypoxic conditioning effects and how these pathways can be targeted for chemical conditioning. Despite the present scarcity of knowledge on the efficacy of such approaches in neurodegeneration, the specific characteristics of Huntington’s disease may make it particularly amenable for such conditioning techniques. Not only do classical features of neurodegenerative diseases like mitochondrial dysfunction, oxidative stress and inflammation support this assumption, but also specific Huntington’s disease characteristics: a relatively young age of neurodegeneration, molecular overlap of related pathologies with hypoxic adaptations and sensitivity to brain hypoxia. The aim of this review is to discuss several molecular pathways in relation to hypoxic adaptations that have potential as drug targets in neurodegenerative diseases. We will extract the relevance for Huntington’s disease from this knowledge base.

## 1. Is There Any Potential of Hypoxic and Chemical Conditioning in Huntington’s Disease?

Huntington’s disease (HD) is a rare dominantly inherited brain disorder, which is characterized by progressive striatal and cortical degeneration with associated motor, cognitive and behavioral disturbance [1,2]. HD is caused by the expansion of a polyglutamine stretch (polyQ, >36 repeats) in the N-terminal region of the protein huntingtin (Htt) [3]. Although the function of Htt is still not fully understood, expansion of the polyQ stretch endows mutant Htt (mHtt) with detrimental properties, damaging both neuronal and non-neuronal cells [3,4,5,6,7]. Furthermore, mHtt interferes with mitochondrial functions, such as oxidative phosphorylation and the mitochondrial import system [8,9].

Deficits in brain oxygen supply and energy metabolism are common features of neurodegenerative diseases, and may precede neurodegeneration and clinical manifestations [10,11,12]; for instance, cerebrovascular pathologies have been considered to be important aspects of Alzheimer’s disease (AD) pathology [13]. Mitochondria are the main cellular consumers of oxygen, and sense and react to dysregulation of the oxygen supply. Inadequate oxygen levels thus result in mitochondrial deficits, such as dysregulation of the redox homeostasis and impaired oxidative phosphorylation.

In HD, mitochondrial dysfunction is considered a central pathology and patient brains exhibit both structural [14] and functional [15] abnormalities of mitochondria. Impaired mitochondrial respiration has been reported specifically for the caudate nucleus [16], and mitochondrial DNA may be particularly prone to hypoxia-induced mutations in the putamen and cortex [17], all among the most vulnerable brain regions in HD [2]. Despite these clear mitochondrial pathologies at late stages of the disease, mitochondrial respirational deficits during HD pathogenesis remain difficult to detect in preclinical models [18,19]. We recently argued that this may be due to a relatively maintained mitochondrial respiratory and metabolic capacity under ex vivo testing conditions but under limiting in vivo provision of cells and tissues with oxygen and substrate deficits may manifest [19]. In line with this assumption, apparent mitochondrial respiratory dysfunction in vivo in an HD mouse model was observed only under metabolic stress [20]. Specific substrate limitation has been linked to mitochondrial deficits in aging [21]. Similarly, HD patient brain cells may experience mitochondrial deficits if their energy demand surpasses a certain threshold [22].

Different ways of enhancing mitochondrial functions represent promising strategies to counteract neurodegeneration [23]. We hypothesize that hypoxic conditioning may be a relatively easy to implement strategy to achieve neuroprotective effects. As opposed to many approaches that target specific mitochondrial functions, hypoxic conditioning rather improves the cellular environment for more efficient mitochondria in general. The elucidation of involved molecular pathways may allow pharmacological fine-tuning, complementation or substitution, for example, targeting hypoxic inducible factor (HIF) pathways, the sphingosine system, N-Methyl-D-aspartate (NMDA) receptors or δ-opioid receptors.

## 2. Hypoxic Conditioning, Mitochondria and Neuroprotection

Low oxygen availability (hypoxia, ischemia) has the potential to induce irreversible damage in the brain and is likely involved in neurodegenerative disease pathogenesis [24], such as AD, Parkinson’s disease (PD) and certain types of epilepsies, in which ischemic/hypoxic conditions during and after seizures are considered as a driving force of neuronal loss [25].

Diseases like chronic obstructive pulmonary disease (COPD) and obstructive sleep apnea syndrome (OSA) induce hypoxic episodes that might aggravate neurodegenerative processes [26]. Consequently, patients of COPD and OSA commonly suffer from brain damage and cognitive deficits [27,28,29]. Cellular hypoxia and respective adaptations play key roles not only directly in neurodegenerative processes and neuroinflammation but also in the pathogenesis of associated risk factors, such as cardiovascular diseases [30].

By contrast, controlled exposure to hypoxia (i.e., hypoxic conditioning) may, at the right doses and administration regimes, increase the resilience of the brain to subsequent insults. Hypoxic conditioning is thought to trigger physiological, cellular and molecular hormetic adaptations that render brain structures less susceptible to damage [31]. Mechanistically, these protective adaptations rely mainly on the cerebral vascularization and circulation [32,33] and mitochondrial function [34]. The neuroprotective potentials of hypoxic conditioning have been repeatedly reported, for example, in ischemic [35] or oxidative injury models [36], as well as in kainic acid-induced excitotoxicity and epilepsy models [37,38,39,40]. The main molecular effectors of these adaptations are molecules of the HIF signaling pathways. Their pharmacological or genetic modulation can mimic, fine-tune or abolish adaptations to hypoxia (see below). These pathways are interconnected with various other mechanisms inducing conditioning effects. Examples of targets of chemical conditioning include the δ-opioid receptor [41,42], NMDA receptor [43] and the sphingosine system signaling pathways (discussed in more detail below). In addition, several inhibitors of mitochondrial components can be used for chemical conditioning [44], rendering tissues more tolerant to subsequent injuries by improving mitochondrial and other cellular functions.

However, there is evidence that hypoxic conditioning is less efficient in aged animals. This may be due to reduced HIF function at an advanced age, as shown in *Caenorhabiditis elegans* and in mice [45]. Accordingly, the beneficial outcomes of hypoxic conditioning have been reported to be reduced [46,47] or delayed [48] in aged rodents. Consequently, hypoxic conditioning is a more promising treatment strategy for HD patients, as the average age of onset of around 45 years [2] is lower than for other neurodegenerative diseases, such as sporadic AD and PD. Yet, hypoxic conditioning still works in older people [49]; the beneficial effects of hypoxic conditioning were recently demonstrated in older patients with mild cognitive impairment, an early stage of dementias such as AD [50,51,52]. These results provide hope that hypoxic conditioning remains a potential therapeutic strategy for neurodegenerative diseases at an older age.

## 3. Overlapping Molecular Pathways of Hypoxic and Chemical Conditioning

Low oxygen levels are obviously dangerous for cells that rely on oxygen for a variety of biochemical reactions. Mitochondria produce most of the chemical energy in almost all eukaryotic cell types and also consume most of the oxygen available to organisms, both mainly attributable to oxidative phosphorylation. In response to low oxygen, adaptive mechanisms support cell survival. These adaptations depend on a variety of factors. First, the hypoxic dose (severity, frequency, intermittence, duration of the hypoxic stimuli) determines not only the harm to the affected cells but also the extent and duration of the adaptations. Second, both vulnerability and response to hypoxia depend on characteristics of the cell (cell type, cell environment, etc.) or organism (e.g., health status, gender, genetic makeup, age).

Cellular adaptations to hypoxia encompass a reduction of oxygen utilization, improvement of the energy status of the cell, structural changes to boost cellular oxygen supply and the prevention of hypoxia-induced cell death. Oxidative phosphorylation is a process strongly affected by reduced oxygen supply and is downregulated following hypoxia exposure, since cells upregulate components of alternative energy production, including glycolytic enzymes and glucose transporters to compensate for the loss of energy [53]. In addition, increased fatty acid biosynthesis in human neuroblastoma cells has been reported in response to hypoxia, possibly to counteract brain lactoacidosis and the loss of reduction potential [54].

Central signaling molecules to effectuate these adaptations are reactive oxygen species (ROS) [55] and HIFs (discussed in the following section) that interact with numerous other signaling cascades (Figure 1). These interactions are still poorly understood but shape the foundation of cellular adaptations to hypoxia. In this review, we focus on the role of a few highly interconnected signaling cascades that share integral functions in hypoxic adaptations and in neurodegenerative diseases, particularly in HD. These cascades, revolving around HIFs and the sphingosine and δ-opioid receptor systems, as well as around *N*-methyl-d-aspartate (NMDA) receptors (NMDARs), are discussed in the following sections.

### 3.1. The HIF Pathway

In normoxic conditions, HIF α-subunits are constantly degraded. If 2-oxoglutarate is available, HIF α-subunits are hydroxylated by prolyl hydroxylases, and marked by van Hippel–Lindau (pVHL) proteins for degradation by the 26 s proteasome [53,56,57]. Besides this mechanism, the factor inhibiting HIF-1 (FIH1) also represses HIF transactivation. In hypoxic conditions, oxygen-regulated α-subunits of HIFs form heterodimers with their constitutive HIF β-subunits and activate transcription by binding to hypoxia-responsive elements (HREs) [58] (Figure 2).

Like HIF-1, its paralogs HIF-2 and HIF-3 also regulate adaptations to changes in oxygen levels. Although HIF-1 and HIF-2 bind the same HREs [59], they still induce different effects, possibly because of paralog-specific co-activators or differential temporal activation patterns [53]. The main co-activators of HIFs are the cAMP response element-binding protein (CREB)-binding protein (CBP) and p300 that contribute to recruiting further co-activators [60]. HIF-1 is thought to be the main effector in adaptations to acute hypoxia and HIF-2 appears to be more predominantly involved in chronic hypoxia adaptations [61]. HIF-3 is a weak activator of HREs and might act as a negative regulator of HIF-1 and 2; its splicing variant inhibitory Per/Arnt/Sim (PAS) domain protein (IPAS) is a dominant negative regulator of the other HIFs [53].

HIFs are regulated by a large number of other mechanisms, including post-translational modifications (e.g., deacetylation by sirtuin 1) and interaction with a multitude of other molecular pathways, such as Wnt, mTOR, PPARγ, Myc and Notch [53]. The pharmacological modulation of HIF pathways is of great interest as a potential treatment strategy in several neurodegenerative diseases [62,63,64,65,66], with much preclinical research having been performed for AD [67] and PD. Early approaches targeting components of HIF pathways in models of PD have been summarized by Youdim and colleagues [68]. Since then, many new reports have been published that demonstrate the wealth of possibilities to modulate HIF-related processes as therapeutic strategies for neurodegenerative diseases. For example, inhibition of prolyl hydroxylases [69] resulting in HIF stabilization or the upregulation of a main downstream process of HIF, glycolysis, by terazosin [70] and, furthermore, upregulation of HIF-1 by albendazole [71], agmantin [72], lactoferrin [73] or deferoxamine [74], were successfully applied in cellular and rodent models of PD.

Despite the theoretical potential of the modulation of HIF pathways specifically in HD (see Section 4), limited related information is available on that topic. An interesting link of HIF pathways to the sphingosine system, which is increasingly acknowledged as a major factor in HD pathogenesis and which is presented next, is emerging.

### 3.2. The Sphingosine System

Sphingolipids represent key components of cell membranes [75]. Their synthetic pathways are tightly regulated and involve the coordinated action of several biosynthetic enzymes (Figure 3) [75,76].

Ceramide is the hub in the synthesis of sphingolipids [77]. It represents a key signaling molecule mediating pro-survival and proliferative effects [78] that are important for the early growth and development of neuronal cells [79,80]. Conversely, its accumulation is detrimental for different cell types, including neurons [81,82,83,84]. Similarly, the sphingolipid metabolite sphingosine-1-phosphate (S1P) regulates several molecular events, essential for brain development and neuronal survival [85,86]. S1P is synthesized by sphingosine kinase-1 and 2 (SPHK1 and 2) and degraded by sphingosine-1-phosphate phosphatase (SGPP) or lyase (SGPL1) (see Figure 3) [87,88]. SPHK1 is usually associated with cell survival [87,88,89], while SPHK2 represents a dual-function enzyme, which may exert both beneficial and toxic effects [90,91,92,93,94]. SGPP and SGPL1 activities are fundamental for the maintenance of the balance between S1P and other sphingolipid intermediates that may regulate cell proliferation, growth and survival [95].

S1P is a potent lipid signaling molecule, which can act both intra- and extracellularly [86,96]. It binds to five known G protein-coupled receptors, S1PR_1_–S1PR_5_, which are expressed in many cell types, including neurons [97,98]. Besides its role in proliferation, migration and apoptotic regulation of cells [99], SIP1 is also involved in responses to hypoxia. It has been shown to protect neonatal cardiac myocytes from hypoxic insult in a mitochondrial ATP-dependent K^+^ (KATP) channel- and proteinase kinase C-dependent manner [100]. SPHK1 (but not 2 in this study) has been reported to be regulated by hypoxia via both HIF-1 and HIF-2 [101]. However, S1P also modulates hypoxia-induced adaptations by activating HIF-1 [102,103,104]. This effect is mediated via Akt/GSK3beta signaling [103]. In human umbilical vein endothelial (HUVEC) cells, S1P has also been shown to be an important regulator of vascular endothelial growth factor (VEGF, a vasculogenesis promoter) [105] and well-known target of HIF signaling.

Treatment with exogenous S1P has been demonstrated to be protective against subsequent hypoxic insults and knockout of SPHK1 caused aggravated hypoxic damage in mouse cardiomyocytes [106,107,108]. Similarly, SPHK2 has been reported to be required for hypoxic conditioning [109]. The benefits conferred by S1P appear to be at least partially mediated by positive effects on mitochondrial integrity [110].

Anelli and colleagues [111] reported the regulation of SPHK1 by both HIF-1 and HIF-2 in U87MG glioma cells. These authors demonstrated that SPHK1 transcription, translation and enzyme activities are upregulated in hypoxic conditions. siRNA-mediated knockdown of HIF-2α abrogated these SPHK1 changes at reduced oxygen levels. Conversely, HIF-1α knockdown resulted in increased HIF-2a expression and the upregulation of SPHK1.

More recently, both extracellular S1P signaling and direct interaction of S1P in the nucleus have been shown to regulate HIF signaling in cellular cancer models. Bouquerel and colleagues demonstrated that S1P, via activation of the extracellular S1P receptor 1 and Akt/GSK3beta, regulates HIF-2α expression [112]. S1P has been reported to form complexes with HIF-1α in the nucleus and to modulate HIF-1α-induced transcription by binding HIF-1α promoter regions and enhancing histone H3 acetylation in different cancer cell lines [113].

In humans, sphingosine metabolism has been demonstrated to be crucially involved in erythrocyte adaptations to high altitude. In healthy lowlanders, increased concentrations of S1P and concurrent elevated SPHK1 activity have been observed at exposure to an altitude of 5260 m, which led to an increased hemoglobin oxygen release capacity and thus may reduce tissue hypoxia [114]. The authors of that study also demonstrated a role of S1P in erythrocyte glycolysis activation in hypoxia [114].

Recent work links the sphingolipid system increasingly to aging and neurodegeneration [115,116], with important physiological functions in neuronal functioning but the potential to wreak havoc in the brain due to aberrant signaling [117]. Sphingolipid signaling in the brain is involved in the regulation of neuronal development and survival, synaptic transmission, autophagy and neuroinflammation. Dysregulated sphingosine signaling therefore can disrupt, e.g., synaptic transmission, long-term potentiation and learning and memory (summarized in [117]). While autophagy is generally required for the disposal of cellular waste products, such as damaged proteins, and as a backup for nutrient supply during times of starvation, its dysregulation can be detrimental and is involved in the pathogenesis of numerous diseases, including HD. For example, SPHK1 has been demonstrated to regulate autophagy in primary neurons leading, to an increased clearance of mutated Htt [118]. Furthermore, sphingolipids are implicated in the activation of glial cells in the brain and proper sphingolipid signaling is necessary for the induction of adequate inflammatory processes [118].

Over recent years, it has been extensively demonstrated that HD is characterized by an early impairment of S1P homeostasis [89,119,120,121]. Defective expression of S1P-metabolizing enzymes, along with the reduction of its bioavailability, has been reported in different HD settings [89,120].

The occurrence of these alterations early in the course of the disease [89,120] suggests S1P as a key contributor to the pathogenesis of HD and potential therapeutic target. In this context, the efficiency of enhancing S1P signaling [122,123] or SPHK1 activity in the R6/2 mouse model of HD was shown [121,122]. Although not investigated, it may be that the beneficial effects S1P exerts in HD disease progression are at least in part due to an activation of protective hypoxia-related pathways.

Besides HIFs and the sphingosine system, opioid systems are predominantly involved in the regulation of hypoxia adaptations and required for hypoxic conditioning, as outlined below.

### 3.3. The δ-Opioid Receptor System

The tolerance effects after repeated exposure to hypoxia have been shown to be blocked using an opioid receptor antagonist, naloxone [124]. Later, it became apparent that the main pathway involved in this effect is δ-opioid receptor signaling. δ-opioid receptor-mediated neuroprotection has been prominently described for hypoxia/ischemia [41,42,125] and may be mediated by beneficial effects on mitochondria [126,127]. Some evidence of the involvement of the δ-opioid receptor system in neuroprotective adaptations in response to hypoxia comes from studies in epilepsy [128,129] and PD models [130].

A characteristic of opioid signaling is a high level of promiscuity among receptors and ligands [129]. Besides the primary endogenous ligands of δ-opioid receptors, the enkephalins, other opioids bind them as well. The main roles of enkephalins in hypoxia-relevant functions are the modulation of neurotransmitter release, calcium regulation, neuroinflammation, neuroprotection, energy metabolism, angiogenesis and others [131]. Their anatomical distribution, furthermore, may indicate a role of enkephalins in the regulation of (systemic) respiration [132], and thus also a potential involvement in adaptations to hypoxia on a systemic level.

With regard to neurodegenerative diseases, δ-opioid receptor signaling in the context of hypoxia has been linked, for example, to PD-related proteotoxicity. One of the pathological hallmarks of PD is the misfolding and aggregation of the protein α-synuclein, leading to Lewy pathology (intracellular inclusions consisting mainly of aggregated α-synuclein and membranous particles) [133]. α-synuclein is upregulated in response to hypoxia [134], which can be prevented by the modulation of the molecular hypoxia response via activation of the δ-opioid receptor [130]. It is unclear whether similar processes occur in HD, however, there is evidence that links the regulation of Htt levels to transcriptional responses to hypoxia [135].

Moreover, hypoxic conditioning has been shown to upregulate both enkephalins and δ-opioid receptors [136,137], probably via HIF-1-dependent mechanisms [138]. This is particularly interesting in light of findings that showed that elevating striatal pre-enkephalin levels improved HD symptoms in the R6/2 mouse model of HD [139].

Other receptors intricately involved in hypoxia adaptations are NMDARs, which are the topic of the next section.

### 3.4. NMDA Conditioning

Glutamate is the major excitatory neurotransmitter in the mammalian brain and a key mediator of intercellular communication, plasticity, development and differentiation of neurons. Under normal physiological conditions, the extracellular concentration of glutamate is maintained in the micromolar range, which is crucial for the excitatory postsynaptic signaling through distinct ionotropic (NMDA, α-amino-3-hydroxy-5-methyl-4-isoxazolepropionic acid (AMPA) and kainate) and metabotropic glutamate receptors.

NMDARs are tetrameric receptors composed of different subunits: GluN1, GluN2 (GluN2A–GluN2D) and GluN3 (GluN3A and GluN3B). Predominantly, NMDARs consist of two GluN1 subunits in combination with at least one GluN2 isoform that is necessary for the functionality of the receptor [140,141]. Each of the different subunits shows distinct brain distributions, kinetic properties and regulation, enabling NMDARs to exhibit differential functionality and pharmacological characteristics determined primarily by their subunit composition [142].

Pioneering studies in the early 1980s already demonstrated the neuroprotective potential of targeting glutamate-mediated signaling [143,144]. More recently, it has been shown that GluN2A-containing NMDARs, which are mostly located within the synapse in the adult brain, are linked to neuroprotection. GluN2B-containing NMDARs are predominantly found extrasynaptically and their activation has been associated with excitotoxicity [145] (Figure 4). Interestingly, HD—like several other neurodegenerative diseases—is characterized by increased detrimental extrasynaptic signaling [146] and NMDARs are thought to be crucially involved in HD neuropathology, possibly by mHtt-induced alterations in NMDAR structure, trafficking and function [147].

Numerous studies demonstrated the involvement of NMDARs in acute neurodegeneration after cerebral ischemia [148,149,150,151]. In contrast, mild neuronal depolarization, stimulating primarily synaptic glutamate receptors before ischemia induction, promoted protection against such insult both in vivo and in vitro [152,153,154,155]. These studies suggest that glutamate receptors may be involved in ischemic preconditioning. Indeed, a non-competitive and activity-dependent antagonist, MK-801, administered during ischemic preconditioning, resulted in the blockade of ischemic tolerance in vivo [156], ex vivo [157] and in vitro [158]. Furthermore, selective blockade of GluN2A-containing receptors increased neurodegeneration after transient global ischemia and abolished the induction of ischemic tolerance, while inhibition of GluN2B-NMDAR attenuated ischemic cell death and enhanced preconditioning-induced neuroprotection [159].

In line with this, subtoxic concentrations of NMDA prevented neuronal death induced by glutamate, NMDA (Chuang et al., 1992; Dickie et al., 1996; Boeck et al., 2005) or oxygen and glucose deprivation (Pringle et al., 1999; Valentim et al., 2003), indicating that mild NMDAR activation triggers a mechanism similar to hypoxic conditioning.

The neuroprotective effect of memantine, which may be due to a selective blockage of extrasynaptic NMDARs, supports the hypothesis of potentially detrimental consequences of extrasynaptic NMDARs [160]. On the other hand, synaptic NMDAR activation has been shown to exert adverse effects, for example, by mediating hypoxia-induced excitotoxicity [160].

Recently, another explanation for the differential effects of synaptic and extrasynaptic NMDARs has been brought forth, based on the observation that extracellular NMDARs associate with transient receptor potential (TRP) melastatin subgroup 4 (TRPM4) proteins to form complexes that mediate excitotoxicity [161]. TRPMs have previously been linked to excitotoxicity [162] and to neuronal death in brain ischemia [163] and anoxia [164].

NMDAR activation induces the rapid release of brain-derived neurotrophic factor (BDNF), which in turn activates its receptor TrkB. The activation of both NMDARs and TrkB triggers the expression of NF-kB, a transcription factor expressed by neurons in response to stress- and injury-related stimuli, including exposure to cytokines, excitotoxicity and oxidative insults. Mattson and colleagues [165] demonstrated that NF-kB activation protects hippocampal neurons against oxidative stress-induced apoptosis, implying that NF-kB can protect neurons under certain conditions.

Another key mediator involved in synaptic NMDAR-dependent neuroprotection is cyclic AMP-responsive element-binding protein (CREB). CREB is a transcription factor induced in response to intracellular calcium elevation regulated by synaptic NMDAR activation [166,167,168]. Pharmacological interventions to ischemia are most effective in the area surrounding the ischemic core where revascularization occurs (the penumbral region). In this region, CREB phosphorylation is strongly upregulated in preconditioned rats [169], suggesting that CREB may be an important player in the conditioning mechanism mediated by NMDARs. Indeed, selective blockade of GluN2A-containing NMDARs suppressed CREB phosphorylation after ischemic preconditioning, whereas GluN2B-NMDAR inhibition had no effect [159]. With respect to the possible downstream genes regulated by CREB, it has been reported that CREB phosphorylation induced by ischemic preconditioning enhances the expression of the anti-apoptotic gene Bcl-2 [170], and that NMDAR inhibition reduces Bcl-2 elevation [171]. Furthermore, CREB is responsible for increased levels of the immediate early gene Cpg15, which is involved in neuroprotection by preventing the activation of caspase pathways. Cpg15 induction can be rescued by selective inhibition of GluN2A-NMDARs. Notably, the major co-activators of HIFs are CREB-binding proteins (see Section 3.1).

Current knowledge suggests that NMDARs play a critical role in conditioning effects and their modulation could improve neuronal survival after a critical ischemic or hypoxic insult. However, the mechanisms providing neuroprotection are still largely unknown and certainly not limited to NMDAR-dependent signaling cascades.

In summary, molecular responses to hypoxia are the result of the interplay of different signaling pathways and regulated by numerous messaging molecules and biochemical adaptations. The coordinated response to hypoxic stress enhances the resilience of the cellular and tissue systems to the perceived or anticipated insult. Therein lies the potential of hypoxic conditioning for brain protection.

## 4. Conditioning Benefits on Mitochondrial Dysfunctions, Oxidative Stress and Neuroinflammation: The Relevance for Huntington’s Disease

Hypoxia is involved in neurodegenerative processes and the neurodegenerative mechanisms following the administration of neurodegeneration-related neurotoxins and of hypoxia/ischemia overlap [172]. This raises the question of whether the modulation of molecular pathways involved in adaptations to hypoxia are also protective in neurodegenerative diseases and, in particular, in HD. As outlined above, the pharmacological modulation of pathways involved in hypoxia adaptation, including HIFs, the sphingosine system, δ-opioid receptors and NMDARs, represents a promising approach against neurodegeneration. Conditioning approaches can be applied to strengthen mitochondrial functions, increase the endogenous oxidative stress defense and reduce neuroinflammation [173], processes that are all central in HD pathogenesis (see below). Such strategies are able to improve brain tissue resilience and thus brain cell health, which has been suggested to be an important feature of future neurodegeneration treatment strategies with greater translational potential than previous strategies developed in pre-clinical models [174]. Furthermore, the relatively early average onset as compared to sporadic neurodegenerative diseases render HD an attractive target for conditioning approaches, particularly if conditioning is confirmed to be less efficient in older persons, as discussed in Section 3.

In addition, HD has been associated with an intriguingly altered sensitivity to hypoxic insults. An increased sensitivity of R6/2 HD model mice to ischemia as compared to wild-type mice has been reported at a young age [175]. While this resistance decreased with age in wild-type animals, ischemic resistance was maintained in R6/2 mice, eventually surpassing wild-type mouse resistance. The authors speculated that this effect may represent compensatory mechanisms in R6/2 mice, resulting in increased resistance; an effect resembling hypoxic conditioning. The striatum, a brain region of high vulnerability in HD, has also been shown to be especially vulnerable to hypoxic insults [176]. The even higher vulnerability of GABAergic projection neurons as compared to cholinergic interneurons in neurodegenerative diseases is mirrored by hypoxic insults as well [177]. These observations might indicate a central role of hypoxia and ischemia in HD pathogenesis and in turn the specific potential of hypoxic conditioning.

### 4.1. Mitochondria and Oxidative Stress

One of the main benefits of hypoxic conditioning is the improvement of mitochondrial integrity and function that is thought to be at the core of conditioning-mediated neuroprotection [34,173,178]. A reduced supply of oxygen results in the HIF-mediated upregulation of mitophagy and decreases in mitochondrial mass and metabolism [178], as well as in a net downregulation of mitochondrial respiration [56]. Mitochondrial complex I inhibition [179], in response to hypoxia, has been suggested to be accompanied by a transient complex II upregulation [180]. In addition, complex IV is remodeled; both hypoxia-inducible gene domain family member 1A (HIGD1A)-mediated activity enhancement [181] and the substitution of complex IV subunit COX4I1 by COX4I2 [182] result in increased complex IV efficiency in response to hypoxia.

While many effects of HIFs on mitochondria are mediated by HIFs’ roles as transcription factors in the nucleus (Figure 2), it was recently demonstrated that upon hypoxic or oxidative stress, HIF-1α can also translocate to mitochondria and that mitochondrial localization of HIF-1α protects against apoptosis [183].

Mitochondrial dysfunction is a general feature of neurodegenerative diseases but is especially strongly linked to HD [184]. The observation that the mitochondrial complex II inhibitor, 3-nitropropionic acid (3-NP), induces HD-like selective striatal (caudate-putamen) neurodegeneration corroborates the central role of mitochondria in HD pathogenesis [185]. It is noteworthy that 3-NP is considered as a hypoxia mimetic [186] and 3-NP conditioning has been reported to overlap in various molecular mechanisms with hypoxic conditioning [187]. Accordingly, 3-NP toxicity—and thus most likely also its conditioning effects—is at least partially mediated by NMDARs [188]. Additionally, the disease-causing mHtt protein interacts with mitochondria and may specifically impair complex II [189]. mHtt has been demonstrated to induce mitochondrial fragmentation [190] and cytoplasmic inclusions of mHtt Exon1 have recently been shown to also recruit mitochondria and cause their fragmentation as well as mitochondrial respiratory abnormalities [191]. Hypoxic conditioning, on the other hand, may counteract mitochondrial fragmentation by enhancing mitochondrial fusion [192].

Other reported mitochondrial deficits include dysfunctional mitochondrial biogenesis due to the transcriptional repression of PPARγ coactivator 1α (PGC-1α), a master regulator of mitochondrial biogenesis and respiration, by mHtt [193]. In addition, impaired calcium handling due to disrupted mitochondria–endoplasmic reticulum contacts was shown in a mouse model of HD [194] and mitochondrial metabolism is impaired even in peripheral cells of patients [195].

Oxidative stress and associated oxidative damage are further hallmarks of neurodegenerative diseases and specifically of HD [196,197]. As major ROS producers, mitochondria are intimately linked to oxidative stress. ROS are also important regulators of cellular adaptations to hypoxia [55] and hypoxic conditioning in turn can induce an upregulation of the antioxidative defense capacity [173,198] and may therefore be protective in HD pathogenesis. A recent study in human renal cell and neuroblastoma-derived cell lines demonstrated the protection from toxins (lidocaine in this study) conferred by HIF-1 [199]. These authors report that different approaches to enhance HIF-1 activity resulted in a protective downregulation of the electron transport system and ROS generation: hypoxia, deletion of VHL or HIF hydroxylase inhibition.

In summary, these findings highlight a great potential of controlled hypoxia interventions or targeted pharmacologic modulation of related pathways to enhance mitochondrial function and cellular antioxidative capacities.

### 4.2. Neuroinflammation

Pathological processes like oxidative stress or transcriptional dysregulation in HD cause immune system activation and inflammation in central and peripheral tissues [2]. In a recent report, Lee and colleagues [200] observed the release of mitochondrial RNA (mtRNA) specifically from striatal spiny projection neurons in HD patients and mouse models of HD. Cytoplasmic mtRNA induces strong innate immune responses [201]. In the transcriptomics study of Lee and colleagues [200], components of the neurotrophin pathway were upregulated and elements of oxidative phosphorylation were downregulated in HD spiny projection neurons (of the striatopallidal “indirect pathway”) as a result.

While hypoxia is a potent inducer of inflammation [202], hypoxic conditioning has anti-inflammatory effects [173] and leads to beneficial adaptations of the innate immune system [203]. Therefore, the benefits of hypoxic conditioning with regard to neuroinflammation likely rely on conditioning’s general neuroprotective potential, but also on the involvement of associated signaling pathways in immune system functions like, for example, that of HIF-1 [102].

Hypoxic conditioning-related δ-opioid receptor activation can mitigate neuroinflammation, as recently reviewed by Chen and colleagues [204]. In that context, it is interesting that, in HD, a reduced enkephalin immunoreactivity and an associated diminished glia- and neuroprotective potential of δ-opioid receptor agonists have been reported [205], suggesting reduced δ-opioid receptor levels.

Taken together, hypoxic and chemical conditioning have the potential to improve several adverse cellular conditions that are centrally involved in the pathogenesis of HD. The next section is dedicated to experimental evidence of the efficiency of conditioning approaches in HD.

## 5. Approaches of Hypoxic and Chemical Conditioning in HD

In order to investigate the current state of knowledge on hypoxic conditioning in HD, a literature search was conducted in PubMed, searching for combinations of the terms “hypoxic” or “hypoxia” and “huntington” or “huntingtin”. Hits until 25.9.2020 were taken into account. The searches yielded 170 hits but revealed little direct evidence of hypoxic conditioning effects in HD. In addition, Google Scholar searches, references from identified reviews and private literature databases were employed to complement the findings.

Several lines of research indicate a potential of chemical conditioning in models of HD, targeting pathways involved in hypoxia adaptation, as depicted in Table 1. The potential of hypoxic conditioning or the pharmacological induction of related signaling pathways, such as HIF-1 in cerebral ischemia and neurodegeneration, has been broadly discussed [65], for example, activation of HIF-1 by HIF hydroxylase inhibition [66,206]. Modulatory effects on proliferation and differentiation in human fetal striatal neuroblasts [207] and increased cell viability in cellular models of HD, induced by 3-NP, have been reported [208,209]. In *Caenorhabditis elegans*, knockdown of the prolyl hydroxylase egl-9 or the pVHL ortholog vhl-1, which both prevent HIF1α degradation, has been shown to decrease proteotoxic stress due to mutated Htt [210]. These interventions had similar effects on amyloid beta, the main component of extracellular plaques characterizing AD and, furthermore, extended the lifespan of the worms [210].

Chemical conditioning with relevance for HD has also been performed with 3-NP. Li and colleagues [211] demonstrated increased hypoxia tolerance in mouse hippocampal slices, if the mice were pre-treated with 3-NP. Although no HD model was employed in this study, it is relevant for HD since 3-NP was used as a conditioning mediator. A more direct link to HD was provided by Skillings and Morton [212], who reduced HD phenotypes in the R6/2 mouse HD model after 3-NP conditioning.

In a model of brain ischemic preconditioning, Sharma and Goyal [213] recently observed protective effects against subsequent systemic 3-NP administration on motor coordination and neurochemical resilience in mice, further strengthening the concept of conditioning efficiency in HD models.

## 6. Conclusions

Hypoxic conditioning benefits mitochondria by ameliorating cellular environments; besides enhancing mitochondrial functioning, it boosts cellular oxidative defenses, oxygen and glucose supply and reduces inflammation [173]. Related approaches represent promising novel treatment strategies in diseases of the central nervous system [214] and in particular for neurodegenerative diseases. Although currently underinvestigated, HD may present a particularly interesting target for hypoxic conditioning, due to the discussed relatively early onset, the molecular overlap of HD-related pathologies and hypoxic conditioning capacities, an assumed particular sensitivity of the brain in HD to hypoxia and the fact that the hypoxia mimetic 3-NP induces HD-like pathologies. The discussed potential HIF-mediated compensation during hypoxia of complex II for reduced complex I activity [180] may be particularly relevant for HD with complex II dysfunction at its core as well. Finally, glycolysis abnormalities in HD patient brains, e.g., elevated lactate levels [2], are another feature that theoretically could be ameliorated with hypoxic conditioning approaches.

Approaches to improve the cellular environment, as can be achieved by hypoxic conditioning, are expected to represent a superior approach to target mitochondrial dysfunction in neurodegeneration [174] as compared to targeting specific downstream consequences of mitochondrial damage, in approaches targeting, for example, ROS or ATP levels in HD [215,216]. Hypoxic conditioning by itself is limited in many regards; age [217] and disease may blunt adaptive physiological and molecular adaptations and the selection of the hypoxic dose is a balancing act due to the danger inherent to severe hypoxia. A better understanding of the molecular foundations of adaptations to hypoxia will ultimately yield optimized protocols of hypoxic conditioning-mediated cellular adaptations—fine-tuned or even substituted pharmacologically. Beneficial hypoxic adaptations might also enable the re-establishment of cellular efficiency in conditions in which cellular capacities naturally deteriorate (e.g., due to age).

An important concern in treating neurodegenerative diseases is timing. Usually, when symptoms occur, a large number of neurons havr already degenerated and the brain environment promotes continuing neurodegeneration. Recently, Motori and colleagues [218] demonstrated that anaplerotic replenishment of tricarboxylic acid (TCA) cycle intermediates has the potential to rescue mitochondrial dysfunctions and reverse neurodegenerative processes even at advanced disease stages. Based on these promising findings, the investigation into whether optimized hypoxic conditioning-mediated brain remodeling also has the capacity to reverse advanced neurodegenerative processes will be of great interest.

## Figures and Tables

**Figure 1 ijms-22-00582-f001:**
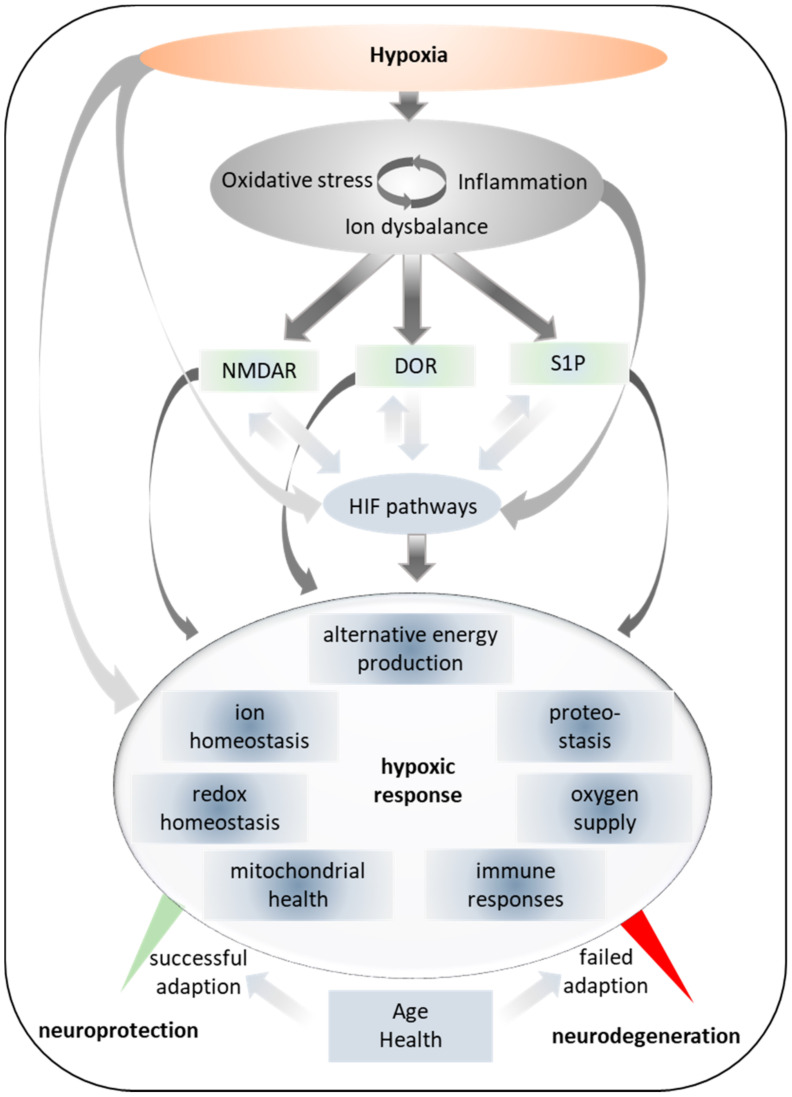
Simplified depiction of selected molecular overlaps in the adaptation to hypoxia. *N*-methyl-d-aspartate (NMDA) receptor—NMDAR; δ-opioid receptor—DOR; S1P—sphingosine-1-phosphate; HIF—hypoxia-inducible factors.

**Figure 2 ijms-22-00582-f002:**
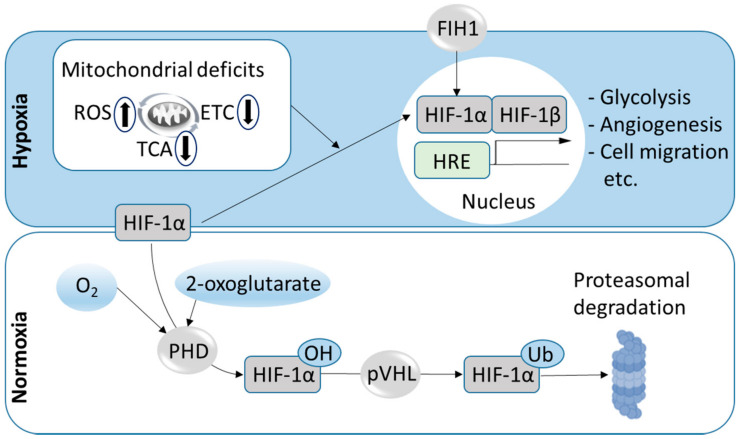
Simplified scheme of HIF-1 regulation. While in conditions of sufficient oxygen (O_2_) and 2-oxoglutarate availability, the α-subunit of hypoxia-inducible factor 1 (HIF-1) is constantly hydroxylated (OH) by prolyl hydroxylases (PHD), which enables recognition by van Hippel–Lindau (pVHL) proteins, ubiquitination (Ub) and proteasomal degradation, HIF-1 α is stabilized during hypoxia, translocates to the nucleus, forms dimers with HIF-1β and binds hypoxia-responsive elements (HREs), triggering transcriptional responses to hypoxia. HIF transactivation is further regulated by the factor inhibiting HIF-1 (FIH1) and consequences of hypoxia on mitochondrial functions, including interdependent effects on the tricarboxylic acid (TCA) cycle, electron transport system (ETS) and mitochondrial reactive oxygen species (ROS) production.

**Figure 3 ijms-22-00582-f003:**
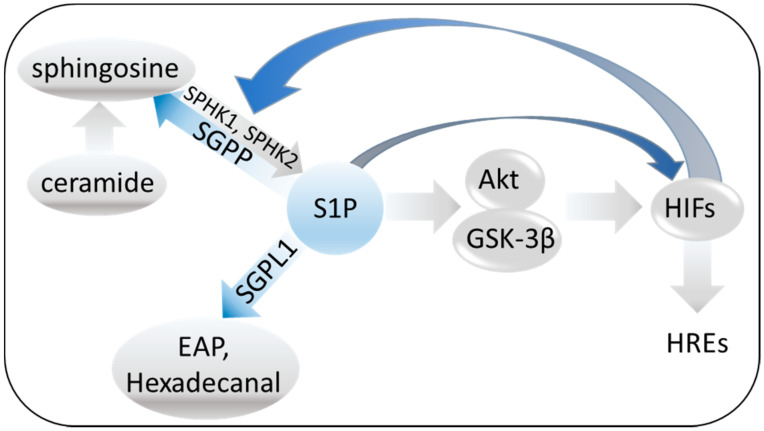
S1P synthesis, degradation and interaction with HIFs. Sphingosine-1-phosphate—S1P; sphingosine kinase-1 and 2—SPHK1 and 2; S1P phosphatase—SGPP; S1P lyase—SGPL1; ethanolamine phosphate—EAP; hypoxia-inducible factors—HIFs, hypoxia response elements—HREs.

**Figure 4 ijms-22-00582-f004:**
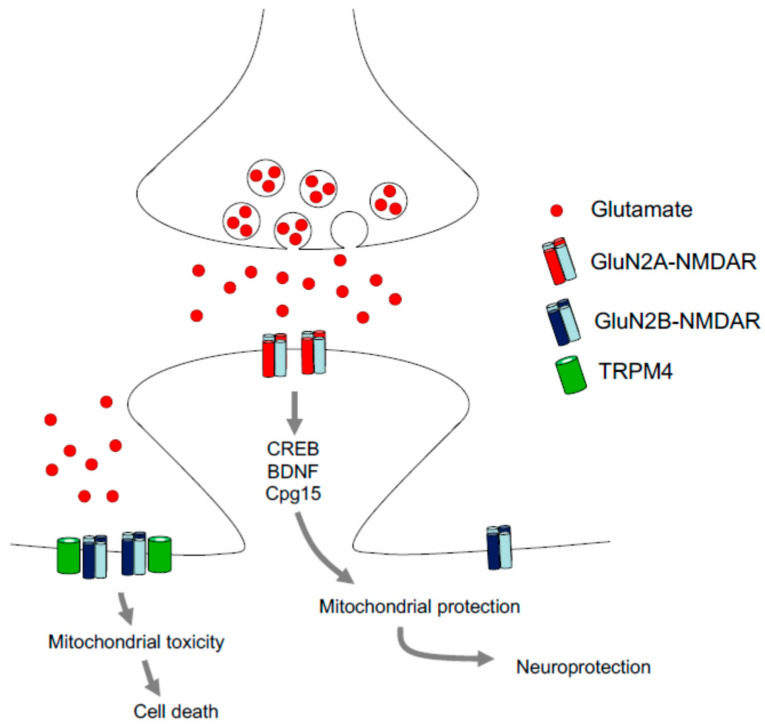
Differential effects of synaptic and extrasynaptic NMDRs. NMDA receptors (NMDARs) can have neuroprotective or detrimental effects, e.g., depending on their localization and subunit composition. Cyclic AMP-responsive element-binding protein—CREB. Brain-derived neurotrophic factor—BDNF; candidate plasticity protein 15 (an immediate early gene)—Cpg15; TRP melastatin subgroup 4 protein—TRPM4.

**Table 1 ijms-22-00582-t001:** Chemical conditioning approaches and Huntington’s disease (HD).

Conditioning Agent	Outcomes	Model	Reference
HIF-1 inducer cobalt chloride (CoCl2)	Hypoxic conditions modulate proliferation and differentiation of human striatal precursor cells and increase VEGF	Human fetal striatal neuroblasts (for transplantation as therapeutic strategy for HD)	[207]
HIF-1 inducers cobalt chloride, mimosine and DFO	Attenuated cytotoxicity	C6 astroglial cells, 3-NP and antimycin A toxicity	[209]
HIF-1 inducer DFO	Attenuated cytotoxicity and increased VEGF, no rescue of complex II deficits	Mouse striatal and cortical neurons, 3-NP toxicity	[208].
HIF-1 stabilization by knockdown of egl-9 or vhl-1	Reduced paralysis and increased lifespan	*Caenorhabditis elegans* with mutated Htt	[210]
3-NP	Increased neuronal hypoxic tolerance	Mouse hippocampal slices, hypoxia	[211]
3-NP	Improvements in multiple behaviors and general health	R6/2 with PolyQ of 250 or 400	[212]

Explanations: hypoxia-inducible factor—HIF; deferoxamine—DFO; 3-Nitropropionic acid—3-NP; vascular endothelial growth factor—VEGF.

## Data Availability

No new data were created or analyzed in this study. Data sharing is not applicable to this article.

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
