# Peer review of "A Rationale for Hypoxic and Chemical Conditioning in Huntington’s Disease"

_ijms, 2021, doi:10.3390/ijms22020582_

Round 1
Reviewer 1 Report
Burtscher and colleagues posit a challenging and intriguing hypothesis that hypoxia and chemical conditioning are potential therapeutic windows for Huntington's disease.
The review covers the essential, necessary information and then proceeded with a thought-provoking idea supported by some research on how hypoxia can be advantageous if we understood the biological mechanisms. Authors cover primary signaling, including crosstalk between sphingosine, SIP, and HIFs pathway and their hypoxia conditionings involvement.
The review is overall well written and is an important contribution to the HD field.
I have the following recommendations.
Potential of hypoxic and chemical conditioning in Huntington's disease?
This question may be rephrased as Is there any potential…..?
In the model, please change SK1, SK2 as SPHK1 and SPHK2 to match the text.
On page 6, Anelli and colleagues….., it is hard to understand what the paragraph conveys? "U87MG glioma cells siRNA mediated downregulation of HIF2α abrogated hypoxia-induced SPHK1 upregulation while targeting of HIF1α with siRNA resulted in increased HIF2a expression and SPHK1 upregulation" This sentence requires rephrasing.
Similarly…" Recent work links the sphingolipid system increasingly to aging and neurodegeneration [109, 110], with important physiological functions in neuronal functioning but the potential to wreak havoc in the brain due to aberrant signaling [111]."
It is useful for the readers to give examples of the critical physiological function and what wreaks havoc in the brain sphingolipids.
Further implicates them in (systemic)… instead of them, please use "these processes… or something similar.
Table 1. c. elegance with mutated Htt
Author Response
Reviewer 1:
Comments and Suggestions for Authors
Burtscher and colleagues posit a challenging and intriguing hypothesis that hypoxia and chemical conditioning are potential therapeutic windows for Huntington's disease.
The review covers the essential, necessary information and then proceeded with a thought-provoking idea supported by some research on how hypoxia can be advantageous if we understood the biological mechanisms. Authors cover primary signaling, including crosstalk between sphingosine, SIP, and HIFs pathway and their hypoxia conditionings involvement.
The review is overall well written and is an important contribution to the HD field.
Re: We are very grateful to the reviewer to acknowledge the importance of our hypothesis for the HD field and are happy about the commendation on both overall writing as well as the thought-provoking potential of our manuscript.
Please note that the indicated line numbers refer to the manuscript version with tracked changes.
I have the following recommendations.
Potential of hypoxic and chemical conditioning in Huntington's disease?
This question may be rephrased as Is there any potential…..?
Re: We agree and changed the section title according to this suggestion (line 38).
- In the model, please change SK1, SK2 as SPHK1 and SPHK2 to match the text.
Re: We thank the reviewer for making us aware of this mistake in Figure 3 (former Figure 2). It has been corrected in the revised version.
- On page 6, Anelli and colleagues….., it is hard to understand what the paragraph conveys? "U87MG glioma cells siRNA mediated downregulation of HIF2α abrogated hypoxia-induced SPHK1 upregulation while targeting of HIF1α with siRNA resulted in increased HIF2a expression and SPHK1 upregulation" This sentence requires rephrasing.
Re: We agree that this sentence was unclear. We modified and clarified as follows (lines 218-222):
Anelli and colleagues [105] reported the regulation of SPHK1 by both HIF1 and HIF2 in U87MG glioma cells. These authors demonstrated that SPHK1 transcription, translation and enzyme activities are upregulated in hypoxic conditions. siRNA mediated knockdown of HIF2α abrogated these SPHK1 changes at reduced oxygen levels. Conversely, HIF1α knockdown resulted in increased HIF2a expression and upregulation of SPHK1.
- Similarly…" Recent work links the sphingolipid system increasingly to aging and neurodegeneration [109, 110], with important physiological functions in neuronal functioning but the potential to wreak havoc in the brain due to aberrant signaling [111]."
It is useful for the readers to give examples of the critical physiological function and what wreaks havoc in the brain sphingolipids.
Re: We are grateful to the reviewer for highlighting this gap in the argumentation. We expanded this section accordingly (lines 237-347):
Sphingolipid signaling in the brain is involved in the regulation of neuronal development and survival, synaptic transmission, autophagy and neuroinflammation. Dysregulated sphingosine signaling therefore can disrupt e.g. synaptic transmission, longterm potentiation and learning and memory (summarized in [111]). While autophagy is generally required for the disposal of cellular waste products, such as damaged proteins, and as a backup for nutrient supply during times of starvation, it’s dysregulation can be detrimental and is involved in the pathogenesis of numerous disease, including HD. For example, SPHK1 has been demonstrated to regulate autophagy in primary neurons leading to an increased clearance of mutated Htt [112]. Furthermore, sphingolipids are implicated in the activation of glial cells in the brain and proper sphingolipid signaling is necessary for the induction of adequate inflammatory processes [112].
- Further implicates them in (systemic)… instead of them, please use "these processes… or something similar.
Re: We thank the reviewer for pointing out this ambiguous formulation. It has been changed as follows (lines 270-272):
Their anatomical distribution furthermore may indicate a role of enkephalins in the regulation in (systemic) respiration [125], and thus also a potential involvement in adaptations to hypoxia on a systemic level.
- Table 1. c. elegance with mutated Htt
Re: Thank you – the typo has been corrected in table 1 and we checked carefully for other typographical errors again.
Reviewer 2 Report
My main concern with the review manuscript by Burtscher at al entitled “A rational for hypoxic and chemical conditioning in Huntington’s disease” is following:
The authors state that they review methods of chemical and hypoxic conditioning strategies, but I do not really find methods and strategic approaches.
One of the focuses of the review is the hypoxic inducible factor (HIF) pathways, therefore in my opinion it would be more beneficial to the readers to introduce the members of the pathway in more details and discuss the role ad possible strategic targets with respect to Huntington’s disease (HD).
The same applies to mitochondria. The chapter of mitochondria is too generalized. The role of mitochondria in HD is well documented in the scientific literature. What do the authors think about possible therapeutic strategies and approaches again in the context with hypoxia?
Author Response
Reviewer 2:
Comments and Suggestions for Authors
Please note that the indicated line numbers refer to the manuscript version with tracked changes.
My main concern with the review manuscript by Burtscher at al entitled “A rational for hypoxic and chemical conditioning in Huntington’s disease” is following:
- The authors state that they review methods of chemical and hypoxic conditioning strategies, but I do not really find methods and strategic approaches.
Re: We thank the reviewer for highlighting this discrepancy. We agree that a more detailed discussion on methods and strategic approaches related to HIF are of interest for the present review. An additional figure has been created to depict the regulation of HIF-1 (figure 2) and the related sections of the review have been expanded (see points 2 and 3).
In addition, we modified the abstract to summarize the manuscript’s contents more accurately (lines 22-25):
We review the involvement of important molecular pathways (e.g. the sphingosine, δ-opioid receptor or NMDA-receptor pathways) in neuroprotective hypoxic conditioning effects and how these pathways can be targeted for chemical conditioning.
- One of the focuses of the review is the hypoxic inducible factor (HIF) pathways, therefore in my opinion it would be more beneficial to the readers to introduce the members of the pathway in more details and discuss the role ad possible strategic targets with respect to Huntington’s disease (HD).
Re: We agree that these methodological aspects with regard to the HIF-pathway were not sufficiently presented. The new figure 2 aims to improve the presentation of the regulation of HIFs. Due to the limited number of relevant publications for HD, in addition, we present several approaches to modulate HIF pathways in PD models now in section 3.1 that may serve as an inspiration for experimentation on HIF in HD models (lines 172-179):
Early approaches targeting components of HIF pathways in models of PD have been summarized by Youdim and colleagues [68]. Since then many new reports have been published that demonstrate the wealth of possibilities to modulate HIF-related processes as therapeutic strategies for neurodegenerative diseases. For example, inhibition of prolyl hydroxylases [69] resulting in HIF-stabilization or the upregulation of a main downstream process of HIF, glycolysis, by terazosin [70] and furthermore upregulation of HIF-1 by albendazole [71], agmantin [72], lactoferrin [73] or deferoxamine [74] were successfully applied in cellular and rodent models of PD.
- The same applies to mitochondria. The chapter of mitochondria is too generalized. The role of mitochondria in HD is well documented in the scientific literature. What do the authors think about possible therapeutic strategies and approaches again in the context with hypoxia?
Re: We thank the reviewer for this suggestion. The section on mitochondria has been re-structured and extended to highlight specific potentials of hypoxic conditioning (lines 397-407).
Reduced supply of oxygen results in HIF-mediated upregulation of mitophagy, decreases in mitochondrial mass and metabolism [178] as well as in a net downregulation of mitochondrial respiration [56]. Mitochondrial complex I inhibition [179] in response to hypoxia has been suggested to be accompanied by a transient complex II upregulation [180]. In addition, complex IV is importantly remodeled; both hypoxia-inducible gene domain family member 1A (HIGD1A) mediated activity enhancement [181] and substitution of complex IV subunit COX4I1 by COX4I2 [182] result in increased complex IV efficiency is in response to hypoxia.
While many effects of HIFs on mitochondria are mediated by HIFs’ roles as transcription factors in the nucleus (figure 2), it recently was demonstrated that upon hypoxic or oxidative stress HIF-1α also can translocate to mitochondria and that mitochondrial localization of HIF-1α protects against apoptosis [183].
Round 2
Reviewer 2 Report
The authors made the suggested changes, therefore I am supportive of publication.